# Global Retrieval of 24-hourly Solar-Induced Chlorophyll Fluorescence and Evapotranspiration from OCO-2, OCO-3 and ECOSTRESS over 1982–2022

Zhuoying Deng<sup>1, 2, #</sup>, Tingyu Li<sup>1, 2, #</sup>, Jinghua Chen<sup>3</sup>, Shaoqiang Wang<sup>1, 2, 3, 4, \*</sup>, Kun Huang<sup>5, 6</sup>, Peng Gu<sup>1, 2</sup>, Haoyu Peng<sup>1, 2</sup>, Zhihui Chen<sup>1, 2</sup>

 <sup>1</sup>Hubei Key Laboratory of Regional Ecology and Environmental Change, School of Geography and Information Engineering, China University of Geosciences, Wuhan, Hubei, China
 <sup>2</sup>Engineering Technology Innovation Center for Intelligent Monitoring and Spatial Regulation of Land Carbon Sinks, Ministry

 of Natural Resources, Wuhan, Hubei, China
 <sup>3</sup>Key Laboratory of Ecosystem Network Observation and Modeling, Institute of Geographic Sciences and Natural Resources Research, CAS, Beijing, China
 <sup>4</sup>College of Resources and Environment, University of Chinese Academy of Sciences, Beijing, China
 <sup>5</sup>Zhejjang Zhoushan Island Ecosystem Observation and Research Station, School of Ecological and Environmental Sciences,

East China Normal University, Shanghai, China

<sup>6</sup>Institute of Eco-Chongming (IEC), East China Normal University, Shanghai, China <sup>#</sup>These authors contributed equally to this work.

\**Correspondence to*: Shaoqiang Wang (sqwang@igsnrr.ac.cn)

**Abstract.** Solar-induced chlorophyll fluorescence (SIF) and evapotranspiration (ET) have been widely recognized as proxies for carbon gain and water loss at the ecosystem level. However, most SIF and ET products on the global scale are generally at

- coarse temporal resolutions of at least one day, which limits their ability to characterize diurnal carbon and water cycles. In this study, we extended the spatiotemporal scale of satellite SIF (from OCO-2 and OCO-3) and ET (from ECOSTRESS) data using machine learning methods, resulting in a global hourly SIF and ET dataset (HOUR\_SIFoco and HOUR\_ET<sub>ECO</sub>) spanning from 1982 to 2022, with a spatial resolution of 0.1°. Our product also provides photosystem-level SIF derived from direct estimation and simulation of Soil Canopy Observation, Photochemistry and Energy fluxes (SCOPE) model, aiming to offer a
- more accurate description of photosynthesis. Our satellite-derived products show good correlations with in-situ flux tower measurements from the FLUXNET2015 community (hourly-scale median R<sup>2</sup> for SIF: 0.72, and ET: 0.53; daily-scale median R<sup>2</sup> for SIF: 0.73, and ET: 0.63). Globally, our product shows good consistency with popular SIF and ET gridded products: the mean proportions of pixels with monthly R<sup>2</sup> exceeding 0.7 are 69.5% and 68.1% when compared with four popular products, respectively. The causal-based attribution analysis revealed significant spatial heterogeneity in the lagged effects of different
- environmental factors on SIF, ET, and water use efficiency based on SIF and ET on the global scale. Overall, our dataset will provide new insights for monitoring the diurnal variations of carbon and water cycles and deepen our understanding of their changes over the past 40 years. The global hourly SIF and ET dataset (1982–2022) at 0.1° spatial resolution produced in this study is available at https://doi.org/10.57760/sciencedb.ecodb.00177 (Deng et al., 2025b).

# **1** Introduction

Carbon and water cycles are the key processes within terrestrial ecosystems (Piao et al., 2020; Regnier et al., 2022). In this cycle, photosynthesis of vegetation forms the gross primary production (GPP), which serves as a major carbon sink in the ecosystem (Miller et al., 2023; Yu et al., 2022); transpiration from vegetation and evaporation from soil and water (referred to as evapotranspiration, ET) is essential for regulating surface energy balance (Cheng et al., 2017; Yang et al., 2023). Therefore, a deeper understanding of photosynthesis and ET is needed especially under global climate change (Rockström et al., 2023; Wankmüller et al., 2024; Zhang K. et al., 2024). However, accurately quantifying these processes on the global scale still poses severe challenges (Fuentes et al., 2024; Lai et al., 2024; Ryu et al., 2019).

In recent years, huge advancements in remote sensing have made it possible to monitor carbon and water cycles on the global scale (Huang et al., 2018; Rodell et al., 2023; Xiao et al., 2019). Solar-induced chlorophyll fluorescence (SIF) and ET are two representative advancements (Xiao et al., 2021). SIF is the faint signal re-emitted by vegetation after absorbing light energy during photosynthesis (Jiao et al., 2019; Porcar-Castell et al., 2021; Van der Tol et al., 2014), which addresses the limitations of traditional optical vegetation indices and has shown great potential in estimating GPP (Sun et al., 2023a), monitoring environmental stress (Mohammed et al., 2019) and plant phenology (Huang et al., 2023). The integration of SIF and ET can further enhance our understanding of plants' functional characteristics, such as analyzing vegetation's water use strategies under various environments through water use efficiency (Zhang Z. et al., 2023a).

Despite numerous studies leveraging satellite-based SIF and ET data for large-scale ecosystem monitoring, data availability remains a major limitation (Elnashar et al., 2021; Sun et al., 2023b). For SIF, this limitation arises primarily from the lack of satellites specifically designed to detect SIF due to its faint signal, resulting in current observations being characterized by coarse spatiotemporal resolution and sparse sampling (Quiros-Vargas et al., 2022). To overcome this, machine learning models and light use efficiency (LUE) models have been developed to upscale low-resolution and temporally discontinuous SIF remote sensing products into high-resolution and spatiotemporally continuous datasets, such as CSIF (Zhang Y. et al., 2018), GOSIF (Li and Xiao, 2019), and SIF005 (Wen et al., 2020). The temporal resolution of these global products typically ranges from 4 days to 1 month, with the highest resolution reaching one day (such as TROPOMI SIF (Lorente et al., 2021)), which can meet

- the commonly need for monitoring seasonal vegetation photosynthesis and estimating GPP. However, these products typically represent fixed-time SIF values, determined by satellite overpass schedules—for instance, around 9:30 AM local time for the MetOp-A satellite (August et al., 2012), and approximately 1:30 PM local time for TROPOMI (Lorente et al., 2021), OCO-2 (Sun et al., 2018), and TanSat satellites (Liu et al., 2018). Most products convert instantaneous SIF values to daily averages using scaling factors such as the ratio of the instantaneous cosine of the solar zenith angle (cos(SZA)) to its daily average (Li
- and Xiao, 2019) or ratios derived from atmospheric radiative transfer models (Zhang Y. et al., 2018). However, all these methods fail to capture the diurnal variations in SIF. Compared to SIF datasets, spatiotemporally expanded ET datasets are

even scarce (Leng et al., 2024), limiting the ability to monitor diurnal changes in water fluxes and the joint application of SIF and ET (e.g. water use efficiency) on the hourly scale.

- Recent satellite missions, such as Orbiting Carbon Observatory-3 (OCO-3) and the Ecosystem Spaceborne Thermal Radiometer Experiment on Space Station (ECOSTRESS) have provided new opportunities to enhance spatiotemporal resolution for SIF and ET (Xiao et al., 2021). OCO-3 and ECOSTRESS, mounted on the International Space Station, enable observations of the same region at different times of the day (Taylor et al., 2020). This unique feature offers the potential to construct diurnal SIF and ET datasets for specific regions (Xiao et al., 2021). Recent studies have utilized OCO-3 SIF and 75 ECOSTRESS ET data to investigate the Amazon rainforest's response to global warming (Zhang Z. et al., 2023a; Zhang Z. et al., 2023b). However, these missions provide SIF and ET data with sparse or coarse spatial sampling, necessitating spatiotemporal upscaling for broader application in terrestrial ecosystem monitoring (Zhang Y. et al., 2023). Recent studies have attempted to produce hourly-scale SIF or ET datasets. However, current methods still face limitations in spatial resolution and extent (0.5° or non-global scale) due to computational power or data availability,
- which restrict their broader and more in-depth applications (Deng et al., 2025a; Jeong et al., 2024; Zhang Z. et al., 2023b). Specifically, producing the dataset at an hourly resolution would lead to an order-of-magnitude increase in computational demand and require more efficient algorithms. In addition, most SIF retrieval methods are based on satellite-observed optical reflectance (from satellites such as MODIS), which limits the temporal range and continuity of the input data, thereby affecting the production of SIF datasets. Finally, some studies have pointed out that satellite-observed SIF may be greatly affected by
- hotspot effects, resulting in substantial measurement biases (Zeng et al., 2023). Converting satellite-derived canopy SIF into photosystem level SIF may help enhance its correlation with GPP (Zhang Z. et al., 2023b), while most SIF datasets do not take this into account.

In this study, we aim to develop a long-term (from 1982 to 2022), high-temporal-resolution (one hour) global SIF and ET dataset with an efficient algorithm based on the OCO-2 and OCO-3 satellites and the ECOSTRESS mission (HOUR\_SIFoco and HOUR\_ET<sub>ECO</sub>). We also attempted to convert satellite-observed SIF into the total SIF emitted by vegetation, to achieve a more accurate modeling of actual photosynthesis. All output data are produced with a spatial resolution of 0.1° and validated by site-level observations and popular regional-scale products. Additionally, we conducted a preliminary long-term analysis to explore how carbon and water fluxes varied and how they were regulated by environmental factors in the past 40 years.

### 95 2 Data and methodology

The technical workflow of this study, as shown in Fig. 1, involves several critical steps. First, raw satellite observation data are collected and pre-processed (Sect. 2.1 to 2.3). Next, photosystem-level SIF is estimated (Sect. 2.4), and the spatiotemporal

Science Science

upscaling model is constructed (Sect. 2.5). Following the production of our dataset, a rigorous validation process is conducted before being used for attribution analysis (Sect. 2.6).