# Peer review of "Global Retrieval of 24-hourly Solar-Induced Chlorophyll Fluorescence and Evapotranspiration from OCO-2, OCO-3 and ECOSTRESS over 1982–2022"

_Earth System Science Data, 2025_

## Referee Comment (RC1)

*Review of Deng et al 2025 by Dr Paul Blackwell Geraldton WA*

**Global Retrieval of 24-hourly Solar-Induced Chlorophyll Fluorescence and Evapotranspiration from OCO-2, OCO-3 and ECOSTRESS over 1982–2022**

Zhuoying Deng1, 2, #, Tingyu Li1, 2, #, Jinghua Chen3, Shaoqiang Wang1, 2, 3, 4, *, Kun Huang5, 6, Peng Gu 1, 2, Haoyu P 5 eng 1, 2, Zhihui Chen 1

*"Conclusion*

*555 In this study, we employed machine learning-based data fusion and scale-up methods to propose, for the first time, a long-term (1982–2022) and high-temporal resolution (1-hour) dataset of SIF and ET with a spatial resolution of 0.1° (HOUR_SIFOCO and HOUR_ETECO). At the same temporal span and resolution, we also provided two comparable photosystem-level SIF datasets using two different methods. We thoroughly evaluated the produced products using both site-level and regional-scale data and demonstrated the high accuracy of these datasets. The long-term analysis highlighted their unique advantages in capturing the560 diurnal dynamics of carbon and water cycles. We also applied an advanced causal analysis method (SURD) to investigate the regulatory effects of four environmental factors (PAR, VPD, soil moisture, and air temperature) on SIF, ET, and water use efficiency (WUE). The results revealed strong spatial clustering and variability on the global scale. In conclusion, our dataset offers great potential for advancing our understanding of terrestrial ecosystem responses to climate change and improving the monitoring of diurnal carbon and water cycles on the global scale."*

This body of work is extremely well composed and analysed, it has a great potential to assist farmers and environmental managers to assess their needs using just remote data, that has been well corroborated with on land met measures.

**However,** and with so many similar studies dominated by remote sensing, there is quite inadequate ground truthing of the data with sound measurements of actual plant growth, water use and yield, as well as actual diurnal canopy temperatures.

Without such reliable ground truthing of such remote sensed data analysis the whole application risks predicting quite different values from reality in selected zones, especially in the deep narrow valleys of parts of the Swiss Alps which are in shade for most of the day and restrict the dairy farmers to short periods of grazing and dominant use of feed in cattle shelters.

I trust that the authors will explain their strategy adequately in a reply and at least recognise the potential shortcoming s of not using actual crop and pasture performance data for ground truthing.

PSB 9/4/2025